# Path Planning Algorithm for Manipulators in Complex Scenes Based on Improved RRT*

**DOI:** 10.3390/s25020328

**Published:** 2025-01-08

**Authors:** Xiqing Zhang, Pengyu Wang, Yongrui Guo, Qianqian Han, Kuoran Zhang

**Affiliations:** 1School of Vehicle and Transportation Engineering, Taiyuan University of Science and Technology, Taiyuan 030024, China; gyr0826@163.com (Y.G.); s202324211138@stu.tyust.edu.cn (Q.H.); 2024017@tyust.edu.cn (K.Z.); 2Smart Transportation Laboratory in Shanxi Province, Taiyuan 030024, China

**Keywords:** six-degree-of-freedom robotic arm, path planning, RRT*, APF, triangular inequalities

## Abstract

Aiming at the problems of a six-degree-of-freedom robotic arm in a three-dimensional multi-obstacle space, such as low sampling efficiency and path search failure, an improved fast extended random tree (RRT*) algorithm for robotic arm path planning method (abbreviated as HP-APF-RRT*) is proposed. The algorithm generates multiple candidate points per iteration, selecting a sampling point probabilistically based on heuristic values, thereby optimizing sampling efficiency and reducing unnecessary nodes. To mitigate increased search times in obstacle-dense areas, an artificial potential field (APF) approach is integrated, establishing gravitational and repulsive fields to guide sampling points around obstacles toward the target. This method enhances path search in complex environments, yielding near-optimal paths. Furthermore, the path is simplified using the triangle inequality, and redundant intermediate nodes are utilized to further refine the path. Finally, the simulation experiment of the improved HP-APF-RRT* is executed on Matlab R2022b and ROS, and the physical experiment is performed on the NZ500-500 robotic arm. The effectiveness and superiority of the improved algorithm are determined by comparing it with the existing algorithms.

## 1. Introduction

The six-degree-of-freedom manipulator plays a crucial role in modern manufacturing, healthcare, service, and logistics, as shown in Figure 1. With the rapid advancement of automation technology, the application scenarios for robotic arms have become increasingly complex, necessitating their ability to perform multiple tasks efficiently and safely in dynamic and uncertain environments [1]. Path planning, as the core problem of robotic arm motion control, aims to generate a feasible path for the robotic arm from the start position to the target position. Simultaneously, it is essential to ensure that the path avoids collisions with obstacles present in the environment throughout the motion [2].

Path planning algorithms can be categorized into two types: search-based path planning algorithms and sampling-based path planning algorithms [3]. Search-based path planning algorithms, such as Dijkstra [4] and the A* algorithm [5], determine the optimal path from a starting point to an endpoint by exploring a discrete state space. They typically operate on a grid or grid graph, constructing a state space graph by connecting adjacent states, and executing pathfinding based on this structure. Although these algorithms demonstrate strong performance in simple environments, they frequently encounter challenges, including low computational efficiency and unsmooth paths in high-dimensional spaces and complex obstacle environments [6]. Unlike search-based algorithms, sampling-based path planning algorithms, such as Rapidly Exploring Random Trees (RRT) [6] and Probabilistic Roadmaps (PRM) [7], generate paths by randomly sampling points within the state space. They are generally regarded as probabilistically complete. If a path exists, the algorithm can certainly identify a feasible route over time. Even if the PRM is theoretically complete, it encounters several challenges in practical applications. These challenges include low efficiency, time-consuming collision detection, a lack of consideration for the robot’s kinematic model, and efficiency losses associated with the two-stage method [8,9]. In addition, the PRM offers probabilistic completeness but does not guarantee optimality. Despite the RRT is effective in navigating high-dimensional spaces, it has several drawbacks. These include significant randomness, slow search speeds in environments with complex obstacles, and the generation of non-optimal planning paths [10,11].

In order to address the issue of the large number of nodes and non-optimal paths generated by the RRT, Karaman et al. [12] proposed the RRT*. The algorithm incorporates a cost function based on RRT. It reduces the randomness of the final path by eliminating the selection of the parent node, which results in asymptotic optimality; however, this approach increases the planning time. Adiyatov et al. [13] proposed a rapid exploration algorithm known as the Randomized Tree Fixed Nodes (RRT*FN). The algorithm restricts the maximum number of nodes, and its methods for sampling, node expansion, and parent selection are consistent with those of the RRT*. Although this method prevents the infinite growth of the tree and conserves memory, it does not significantly enhance planning speed, and the convergence accuracy remains low. To address the issue of the prolonged execution time of the RRT*, Lavalle et al. [14] proposed the bidirectional extended randomized tree algorithm, Bi-RRT. The algorithm enhances pathfinding by extending it from both directions, significantly improving the efficiency of search and path planning. Zhou et al. [15] developed a novel strategy for searching nearest neighbors, known as the generalized distance method, which significantly reduces search time. Aiming at the issue of blind node sampling in the RRT* algorithm, Yi et al. [16] proposed an enhanced version known as the improved P-RRT* algorithm based on RRT* (improved P-RRT*). The enhanced P-RRT* employs two expansion methods for generating new nodes, thereby improving the algorithm’s search efficiency. The first extension employs a target bias extension strategy. The second extension utilizes a random sampling strategy within rectangular regions. Fazhan Tao et al. [17] influenced the generation of random sampling points by introducing a variable probability target bias strategy. Combining enhanced artificial potential field methods with growing trees significantly improves the search speed and path quality of randomized trees and reduces the generation of invalid nodes. Wang et al. [18] utilized Gaussian mixed regression to identify key features in human demonstrations, thereby creating a probability density distribution of human trajectories. It is used to guide the sampling process, enabling the rapid generation of feasible paths. In light of the issue of slow search speeds in narrow regions, Wang et al. [19] proposed the NRRT*. The methodology utilizes the optimal paths derived from the A* algorithm as the training set. It guides the expansion and path search processes that are enhanced by leveraging the predictive and optimization capabilities of neural networks, which facilitate a more efficient sampling process in narrow spaces. Q.C. [20] proposed the RJ-RRT. It employs a novel greedy sampling space reduction strategy along with an environmental assessment approach. The method reduces redundant nodes while accelerating the expansion of the random tree toward the target region. It can identify narrow passages and utilize subtrees to explore these passages internally.

In the field of path planning, while the traditional RRT algorithm performs effectively in many scenarios, one of its major drawbacks is its insufficient consideration of the impact of obstacles on path planning. Khatib et al. [21] first proposed the APF algorithm in 1986. The algorithm demonstrates strong real-time performance and effectively addresses the obstacle avoidance problem. The core concept of the APF algorithm is to create a virtual force field in Cartesian space. This field incorporates the attractive force exerted by the target point on the robot arm, as well as the repulsive force generated by obstacles in relation to the robot arm. In recent years, researchers have enhanced the artificial potential field method. Wu et al. [22] combined the adaptive dynamic window method with an improved artificial potential field method to enhance the obstacle avoidance capabilities and path planning efficiency of robots operating in complex environments. This was achieved by refining the repulsion function and incorporating dynamic windows. Aiming to address the local minimum problem in traditional path planning, Zhang et al. [23] proposed an enhanced artificial potential field method that incorporates relative velocity to improve the obstacle avoidance capabilities of robots operating in uncertain and complex mobile environments. For the obstacle avoidance problem of the robotic arm, Wang et al. [24] address the issues of the robotic arm easily falling into dangerous areas and experiencing unsmooth paths during the planning process by introducing the Jumping Point Search algorithm and the cubic uniform B-spline function. Aiming to address the issue of robot path planning in dynamic environments, Bounini et al. [25] solved the problem of robots easily falling into local minima and experiencing oscillations by introducing virtual obstacle points and dynamically adjusting potential field parameters.

This study synthesizes the advantages of two algorithms to propose an RRT* robotic arm path planning algorithm that integrates heuristic probabilistic sampling with the artificial potential field method. The heuristic probabilistic sampling strategy is implemented to improve sampling efficiency. The artificial potential field method is introduced to guide the sampling points in avoiding obstacles and moving toward the target point. Constructing gravitational and repulsive fields solves the challenge of pathfinding in complex environments. The triangle inequality is utilized to simplify the redundant intermediate nodes and optimize the path. Ultimately, the effectiveness, reliability, and superiority of the improved algorithm are validated through simulations and physical experiments.

The main innovations of this paper are the following:Candidate Point Probability Calculation in the RRT* [26]: In three-dimensional Cartesian space, the line segment connecting the starting and ending points represents the shortest path. Define a spherical region based on the shortest path as the diameter. All candidate points will be sampled within this sphere. Calculate the distance from the candidate point to the nearest point on the shortest path and direct it towards the target point. By constructing a heuristic function, the weight of each candidate point is evaluated. Based on the weight values, the sampling probability of each candidate point is calculated to guide the algorithm in exploring the search space more efficiently.Obstacle Avoidance Strategy Based on the APF [27]: The search time is significantly increased because the sampled nodes may be located in areas with dense obstacles. APF is introduced to guide the sampling point to avoid obstacles and move toward the target point by constructing a gravitational field and a repulsive field. This method enhances the density of sampling points, alters the direction of node expansion, and addresses the challenge of pathfinding in complex environments.Optimization of Paths Based on Trigonometric Inequalities [28]: The triangle inequality principle is employed to optimize each node along the generated path. The evaluation function is designed to re-evaluate and select the parent node for each node, eliminate redundant nodes along the path, simplify the path, and enhance both the efficiency and clarity of the final path.

The remainder of this paper is organized as follows: Section 2 introduces the improved HP-APF-RRT* in this study and offers a detailed explanation of the three key improvements. Section 3 presents a comparative analysis of the proposed algorithm against traditional RRT, RRT*, P-RRT*, and HP-RRT* through both simulation and physical experiments to validate the effectiveness of the proposed approach. Finally, Section 4 provides a summary and discusses potential avenues for future enhancements.

## 2. Improved RRT* with Heuristic Probability Sampling and APF

### 2.1. Kinematic Modeling and Solution of Robotic Arms

Robotic arms can only move within a defined spatial range due to the interrelated constraints and connections among their numerous joints. This limitation restricts the flexibility and diversity of motion, making it challenging for traditional path planning methods to adequately address the requirements of obstacle avoidance in manipulator path planning [29,30]. This study focuses on a six-degree-of-freedom robotic arm. The arm comprises six rotary joints of varying lengths that are connected in series, creating a modular structure. The 3D model, structural diagram of the robotic arm, and the D-H coordinate system are illustrated in Figure 2. In this structure, the transformation matrix between two adjacent joints is first calculated, and the positional relationship from the base to the end effector is established through multiple matrix operations. Next, the standard Denavit–Hartenberg (D–H) model [31] is employed to establish the configuration of the coordinate system for each connecting rod. Specific D–H parameters are presented in Table 1.

As shown in Table 1, i represents the linkage serial number; $a_{i}$ denotes the shortest distance between the two joint axes; $\alpha_{i}$ indicates the clamp between the two linkages; $d_{i}$ represents the distance along the z-axis from the origin of the i−1 linkage to the origin of the i linkage; $\theta_{i}$ denotes the angle of rotation of the i−1 linkage, which is necessary for the i−1 linkage to align with the i linkage; $\beta_{i}$ specifies the limit of the i-joint.

Motion planning for robotic arms is a multi-body system problem that involves complex dynamics and geometry [32]. According to the algorithm proposed herein, the robotic arm can accurately navigate to a designated workspace location based on previously calculated position information of the singular value point. It can also determine the corresponding angular configurations of the joints at the singular value location, thereby enabling the robotic arm to move directly to the singular value point to perform a specific task or to avoid potential collisions. The motion planning of a robotic arm involves the coordinated movement of multiple rigid structures. By applying the chi-square transformation matrix to each joint coordinate system, the position and orientation of each joint in the robotic arm are calculated. In this article, the enhanced RRT* algorithm is employed to sample and plan the joint space of the manipulator. Collision detection is conducted on each of the generated joint poses to create an effective obstacle avoidance path.

### 2.2. Obstacle Collision Detection

In the manipulator’s workspace, there are both regular and irregular obstacles. The computational complexity of collision detection is high. To reduce computational complexity, the robot arm’s linkage can be abstracted as a cylinder. Therefore, it is essential to assess the potential collisions between each cylindrical feature of the manipulator and every obstacle in the environment. This paper employs the technique of enveloping the obstacle ball to streamline the handling of irregular obstacles. In this manner, the task of collision detection between the manipulator and the obstacle is converted into the problem of calculating the distance between a cylinder and a sphere. By equating the radius of a cylinder to the radius of a sphere, it is only necessary to calculate the distance between a straight line in space and the sphere. The schematic is presented in Figure 3. In addition, the article applies Obstacle Inflation for collision detection with standard obstacles. Consider the radius of a cylinder as the expansion dimension of a standard obstacle, also known as the safety margin. The collision detection between the manipulator and the obstacle is transformed into spatial straight-line collision detection, taking into account the expansion of the object [33]. The schematic diagram is shown in Figure 4.

#### 2.2.1. Collision Detection of Irregular Obstacles

In Figure 3a, $r_{1}$ represents the radius of the robotic arm linkage, $r_{2}$ denotes the radius of the obstacle envelope, and r indicates the distance from the robotic arm linkage to the obstacle. In Figure 3b, the connecting rod of the manipulator is represented as a straight line in space, with its radius equal to that of the obstacle.

For the simplified obstacle collision detection model, it is necessary to calculate only the distance from the straight line to the center of the obstacle in space. The angles of each joint of the manipulator can be determined by solving the inverse kinematics for the desired end position and orientation. According to forward kinematics, the homogeneous transformation matrix for each link’s coordinate system is derived. Therefore, the equivalent space linear equation of each link can be obtained. Because the base of the manipulator is fixed, only five links are required to determine whether the manipulator collides with obstacles. The distance r from the straight line to the center of the obstacle in space is calculated. If r is less than or equal to the sum of the link radius $r_{1}$ and the obstacle envelope radius $r_{2}$, a collision occurs; conversely, if r is greater than this sum, no collision occurs.(1)$$\begin{array}{c} \left\{\begin{array}{c} r\leq r_{1}+r_{2}\\ r>r_{1}+r_{2} \end{array}\right. \end{array}$$

#### 2.2.2. Collision Detection of Regular Obstacles

In Figure 4a, r represents the radius of the robotic arm linkage; d indicates the distance from the robotic arm linkage to the obstacle. In Figure 4b, the linkage of the robotic arm is represented as a simplified spatial straight line $\dot{\vec{l}}=\dot{\vec{p_{2}}}-\dot{\vec{p_{1}}}$. $\dot{\vec{p_{1}}}$ and $\dot{\vec{p_{2}}}$ are the starting and ending points of the linkage, respectively. The standard obstacle uses a cuboid as an example to expand the obstacle and create a larger boundary area, thereby simplifying collision detection. For each link segment, verify whether it intersects with the expanded area of the obstacle. If an intersection point exists, it can be concluded that the robotic arm will collide with the obstacle. If there is no intersection between the connecting rod segment and the obstacle, the minimum distance $d_{\min}$ between the connecting rod segment and the line segment that forms the obstacle surface is calculated to assess the safe distance between the manipulator and the obstacle. The $d_{\min}$ can be expressed as:(2)$$d_{\min}=\min_{\vec{P}\in \dot{\vec{l}},\vec{Q}\in \dot{Obstacle_{face}}}\left|\left|\vec{P}-\vec{Q}\right|\right|$$
where $\vec{P}$ is a point on the robotic arm linkage, denoted as $\vec{P}=\left(x_{P},y_{P},z_{P}\right)$; $\vec{Q}=\left(x_{Q},y_{Q},z_{Q}\right)$ is a point that represents the obstacle surface; $\left|\left|\vec{P}-\vec{Q}\right|\right|=\sqrt{{\left(x_{P}-x_{Q}\right)}^{2}+{\left(y_{P}-y_{Q}\right)}^{2}+{\left(z_{P}-z_{Q}\right)}^{2}}$ denotes the Euclidean distance between points $\vec{P}$ and $\vec{Q}$.

Collision detection for square obstacles is accomplished by examining the intersection of the connecting rod with each face of the square. Each surface of the cube can be represented by a plane equation of the form: ax+by+cz+d=0. *a*, *b*, and *c* are normal vectors to the plane, while d is a constant term associated with the plane. The parameter t for the intersection of the connecting rod with the square face can be expressed as:(3)$$t=\frac{ax_{p_{1}}+by_{p_{1}}+cz_{p_{1}}+d}{a\left(x_{p_{2}}-x_{p_{1}}\right)+b\left(y_{p_{2}}-y_{p_{1}}\right)+c\left(z_{p_{2}}-z_{p_{1}}\right)}$$

If t is within the range of [0, 1], the connecting rod intersects the surface and a collision occurs. Otherwise, no collision takes place.

### 2.3. Improved RRT* Based on Heuristic Probabilistic Sampling

#### 2.3.1. Basics of the RRT*

The RRT* is a deterministic sampling-based path planning approach. It constructs a tree structure within the configuration space through an iterative process to identify the optimal path from the initial node $S_{init}$ to the target node $S_{goal}$. The principle of the RRT* is illustrated in Figure 5. In each iteration, the point $S_{rand}$ is randomly sampled from the free space, and then the node $S_{nearest}$ with the minimum Euclidean distance to $S_{rand}$ is selected from the tree. Based on this node, the step is extended in the direction of $S_{rand}$ to generate a new node, $S_{new}$. A tree node whose distance from $S_{new}$ is less than the sampling radius $r_{near}$ is identified from the tree to create a set $S_{near}$. Then, the parent node is re-selected for $S_{new}$ in the set $S_{near}$ to minimize the path cost from the initial node $S_{init}$ to $S_{new}$, and $S_{new}$ is inserted into the tree. After that, rewrite the nodes in $S_{near}$, excluding the parent of $S_{new}$. First, calculate the total cumulative cost of paths from $S_{init}$ to $S_{new\_new}=\left\{S\in \left(S_{near}-S_{near\_parent}\right)\right\}$ for all nodes. Then, evaluate whether designating $S_{new}$ as a parent node reduces the total cost of paths from $S_{init}$ to $S_{near}\in S_{new\_new}$. If possible, update the parent of these nodes to $S_{new}$. By means of continuous iteration, the RRT guarantees the asymptotic optimality of path cost, thereby facilitating efficient and robust path planning in complex environments.

#### 2.3.2. Improved RRT* Algorithm 

In complex three-dimensional multi-obstacle environments, traditional RRT* encounters challenges related to inefficient search processes and excessive randomness in the selection of sampling points. So as to address the issue of reduced node sampling efficiency in three-dimensional multi-obstacle spaces, this study proposes an improved RRT* algorithm based on heuristic probabilistic sampling (HP-RRT*). The sampling process is optimized by generating a set of candidate nodes within a predefined sampling space and assessing the sampling priority of each candidate node through a specific heuristic function. By employing this method, it is possible to minimize redundant sampling nodes and enhance the efficiency of node sampling, thereby reducing the overall computation time.

As illustrated in Figure 6, the starting point a and the endpoint b of the path are first connected. This line segment serves as the diameter to define a spherical space within which the sampling is conducted. Multiple candidate points are randomly generated in the spherical space as potential path nodes. The line segment connecting the starting point and the endpoint represents the shortest path. Give priority to the point nearest to the shortest path and the point closest to the endpoint. Design the corresponding heuristic function and calculate the weight value for each candidate point. Finally, the sampling probability for each candidate point is calculated and samples are drawn based on the corresponding weight values. The specific realization process is outlined as follows:

Initialization phase;

Starting point: denoted by the symbol a. It serves as the beginning of the search process and represents the starting position for all potential paths.

Goal Point: denoted by the symbol b. It indicates the end point of the search process and serves as the destination for all paths.

Shortest path: In geometric path planning, the straight line segment connecting the start point a and the end point b is regarded as an approximation of the shortest path.

Sphere space: the sphere space is defined by the diameter of the line segment ab as the sampling region. The spherical space encompasses all possible path points, with boundaries defined by the endpoints of the line segment ab.

2.Candidate point generation;

Randomly generated point sets in spherical space are utilized as potential nodes in the search process for subsequent pathfinding. Points are generated based on a probability distribution, ensuring coverage of the entire search space and increasing the likelihood of discovering optimal paths.

3.Heuristic function design;

Heuristic function: provides a thorough assessment for estimating the cost of the shortest path from the current candidate point to the goal point. It considers both the distance from the candidate point to the shortest path and the distance from the candidate point to the goal point. It can be expressed as:(4)$$f\left(n\right)={\left(1-\frac{dist\_to\_line}{radius}\right)}^{weight1}\times {\left(1-\frac{dist\_to\_b}{\left|\left|b-a\right|\right|+radius}\right)}^{weight2}$$
where n denotes the *n*-th candidate point; dist_to_line represents the distance from the candidate point to the line segment ab; radius identifies the radius of the sphere; weight1 controls the extent to which the distance from the shortest line segment influences the weight; dist_to_b expresses the distance from the candidate point to the goal point b; weight2 the extent to which the distance from point b affects the weight.

In order to more accurately capture the nonlinear relationship between weights and distance, a nonlinear function is employed instead of the traditional linear decay model, represented by the following expression.(5)$$f\left(n\right)=e^{-\alpha \frac{dist\_to\_line}{radius}}\times e^{-\beta \frac{dist\_to\_b}{\left|\left|b-a\right|\right|+radius}}$$
where *α* and *β* are parameters that determine the decay rate.

To prevent numerical issues, establish a minimum value to ensure that the weights remain sufficiently large.(6)$$f\left(n\right)=\max\left(f\left(n\right),\delta \right)$$
where δ is a predetermined, very small positive number.

4.Sampling and path selection.

The sampling probability $p_{i}$ for each candidate point is determined by Equation (7).(7)$$p_{i}=\frac{f_{i}}{\sum_{1}^{n}f_{i}}\;\;\;i=1,2,3\ldots n$$
where $f_{i}$ represents the heuristic value of the *i*-th candidate point, and n denotes the total number of candidate points.

### 2.4. Improved HP-RRT* Incorporating the APF

When path planning is conducted within the manipulator’s workspace, it is inevitably influenced by obstacles, thereby increasing the complexity of the path planning process.

In order to solve this problem, based on the kinematic analysis of the robotic arm, the APF is employed for path planning. This approach is integrated with the previously proposed highly efficient probabilistic sampling algorithm, HP-RRT*. In this algorithm, the potential field force of APF is utilized as an additional guiding force to influence the generation of sampling points and the selection of paths. This approach aims to enhance obstacle avoidance and results in a sub-optimal route that is close to the shortest path.

#### 2.4.1. Traditional APF

The robotic arm obstacle avoidance model utilizing the APF algorithm is illustrated in Figure 7. The essence of the algorithm involves meticulously creating a virtual potential field within the workspace of the robotic arm and solving for it. The model is based on two primary potential fields: the target potential field and the obstacle potential field. The target point generates a global gravitational potential field at the end effector position of the manipulator, simulating an attractive force that encourages the manipulator to move toward the target point. On the contrary, the obstacle creates a localized repulsive potential field at the end effector position of the manipulator, simulating a repulsive force that helps the manipulator avoid collisions with the obstacle [34].

When the obstacle is outside the repulsive potential field, the end effector of the manipulator is influenced solely by the gravitational potential field and moves directly toward the target point. However, when the obstacle enters the repulsive potential field, the end of the manipulator is affected by both the repulsive potential field and the gravitational potential field. This superimposed effect of forces enables the robotic arm to dynamically plan a path for obstacle avoidance, effectively avoid obstacles, and continue toward the target point.

In the spatial model of the APF, the target point is strategically identified as the lowest point in the potential field, specifically the minimum point of the potential function. By calculating the potential functions of the gravitational field and the repulsive field, and then superimposing them, one can obtain the potential function of the combined field. Under the influence of the combined potential field, the manipulator moves in the direction of the gradient descent of the potential function until it attains the target state.

The gravitational potential field is primarily associated with the distance between the end position of the robotic arm and the target point. As the distance increases, the value of the potential energy also increases; conversely, as the distance decreases, the value of the potential energy diminishes. The expression for the gravitational potential field is as follows:(8)$$U_{att}\left(C_{goal}\right)=\frac{1}{2}K_{a}p^{2}\left(C_{arm},C_{goal}\right)$$
where $U_{att}\left(C_{goal}\right)$ is the gravitational potential field at the target point; $K_{a}$ is the positively proportional gain coefficient; $C_{arm}$ denotes the current position of the robotic arm end-effector; $C_{goal}$ indicates the desired position of the end-effector in the target configuration; $p\left(C_{arm},C_{goal}\right)$ vector function, the difference between the position vector of the robotic arm’s end-effector, $C_{arm}$, and the ideal position of the target point, $C_{goal}$.

The corresponding gravitational force $F_{att}\left(C_{goal}\right)$ is the negative gradient of the gravitational field. It represents the fastest-changing direction of the gravitational potential field function $U_{att}\left(C_{goal}\right)$. The expression is presented below.(9)$$F_{att}\left(C_{goal}\right)=-\nabla U_{att}\left(C_{goal}\right)=K_{a}p\left(C_{arm},C_{goal}\right)$$

The primary influence of the repulsive potential field is the distance between the robotic arm and the obstacle. The expression for the repulsive potential field is as follows:(10)$$U_{req}\left(C_{obstacle}\right)=\left\{\begin{array}{c} \frac{1}{2}K_{r}{\left(\frac{1}{p\left(C_{arm},C_{obstacle}\right)}-\frac{1}{P_{obstacle}}\right)}^{2}\;,0\leq p\left(C_{arm},C_{obstacle}\right)\leq P_{obstacle}\\ 0,p\left(C_{arm},C_{obstacle}\right)>P_{obstacle} \end{array}\right.$$
where $U_{req}\left(C_{obstacle}\right)$ represents the repulsive potential field; $K_{r}$ is the positive proportionality gain coefficient; $p\left(C_{arm},C_{obstacle}\right)$ is a vector function, the difference between the position vector of the end-effector of the robotic arm, $C_{arm}$, and the obstacle, $C_{obstacle}$; $P_{obstacle}$ represents the maximum influence range of the obstacle on the manipulator.

The repulsive potential field differs from the gravitational potential field in that the robotic arm is not always subjected to the repulsive force exerted by the barrier. When the relative distance between the manipulator and the obstacle exceeds $P_{obstacle}$, it is concluded that the obstacle does not affect the manipulator’s operation. The smaller the relative distance between the arm and the obstacle, the greater the effect of the repulsive force and the higher the potential energy. Conversely, as the relative distance between the arm and the obstacle increases, the effect of the repulsive force diminishes, resulting in a decrease in potential energy.

The corresponding repulsive force is the negative gradient of the repulsive potential field. The expression can be represented as follows:(11)$$\begin{array}{c} F_{req}\left(C_{obstacle}\right)=\left\{\begin{array}{c} K_{r}\left(\frac{1}{p\left(C_{arm},C_{obstacle}\right)}-\frac{1}{P_{obstacle}}\right)\frac{1}{P^{2}\left(p\left(C_{arm},C_{obstacle}\right)\right)}\;,0\leq p\left(C_{arm},C_{obstacle}\right)\leq P_{obstacle}\\ 0,p\left(C_{arm},C_{obstacle}\right)>P_{obstacle} \end{array}\right. \end{array}$$

The magnitude of the robot’s combined potential field is the sum of its repulsive and gravitational potential fields. Therefore, the expression for the total function of the combined potential field is:(12)$$U\left(C_{arm}\right)=U_{att}\left(C_{goal}\right)+U_{req}\left(C_{obstacle}\right)$$

The resultant force expression is:(13)$$F\left(C_{arm}\right)=-\nabla U\left(C_{arm}\right)=F_{att}\left(C_{goal}\right)+F_{req}\left(C_{obstacle}\right)$$

Under the guidance of the virtual potential field, the robotic arm moves toward the direction of the steepest decrease in the combined potential field $U\left(C_{arm}\right)$. This is typically where the final desired goal is situated.

In the application of the traditional APF, issues such as local minima and target unreachability arise [35]. Specifically, the trajectory of the robotic arm can be directed toward a region of force equilibrium, known as the potential well. As shown in Figure 8, the combined force acting on the robotic arm in this region approaches zero. In this state, the manipulator is prone to becoming trapped in a local minimum area, so it cannot achieve the arrival of the target point. Due to the presence of a strong repulsive force field, the manipulator may display oscillatory behavior. This further hinders its ability to reach the intended target location and may ultimately result in the failure of the path planning task.

#### 2.4.2. Improved APF

To address these issues, this paper presents an enhanced gravitational potential field function. This improved method adjusts to changes in the distance between the manipulator and the target point, reduces the stagnation time of the manipulator within the potential well, and improves its ability to escape from local minima. The gravitational potential field can be expressed as:(14)$$U_{att}\left(C_{goal}\right)=\left\{\begin{array}{c} \frac{1}{2}K_{a}p^{2}\left(C_{arm},C_{goal}\right),p\left(C_{arm},C_{obstacle}\right)>P_{g}\\ sK_{a}p\left(C_{arm},C_{goal}\right),p\left(C_{arm},C_{obstacle}\right)\leq P_{g} \end{array}\right.$$
where $P_{g}$ is the environmentally determined distance constant, and s is the constant factor.

When the position of the manipulator relative to the goal point is $p\left(C_{arm},C_{obstacle}\right)\leq P_{g}$, the gravitational force is maintained at a constant value. This ensures that the gravitational force is not too weak as the manipulator approaches the target point, effectively guiding it to move toward the target.

The corresponding expression for the gravitational force, $F_{att}\left(C_{goal}\right)$, is presented in Equation (15).(15)$$F_{att}\left(C_{goal}\right)=\left\{\begin{array}{c} K_{a}p\left(C_{arm},C_{goal}\right),p\left(C_{arm},C_{obstacle}\right)>P_{g}\\ sK_{a},p\left(C_{arm},C_{obstacle}\right)\leq P_{g} \end{array}\right.$$

The construction of the enhanced repulsive potential field is more complex than that of the attractive potential field. By adjusting the influence range and strength of the repulsive field function, the target unreachability can be mitigated, enabling the robotic arm to navigate around obstacles more effectively. The repulsive potential field can be expressed as:(16)$$U_{req}\left(C_{obstacle}\right)=\left\{\begin{array}{c} \frac{1}{2}K_{r}{\left(\frac{1}{p\left(C_{arm},C_{obstacle}\right)}-\frac{1}{P_{obstacle}}\right)}^{2}\;{p_{r}}^{n},0\leq p\left(C_{arm},C_{obstacle}\right)\leq P_{obstacle}\\ 0,p\left(C_{arm},C_{obstacle}\right)>P_{obstacle} \end{array}\right.$$
where $p_{r}$ is the adjustment factor for the distance between the end actuator of the robotic arm and the target point, and n represents the number of normal.

The corresponding repulsive force $F_{req}\left(C_{obstacle}\right)$ consists of two components: $F_{req1}\left(C_{obstacle}\right)$ and $F_{req2}\left(C_{goal}\right)$. Their expressions are as follows:(17)$$F_{req}\left(C_{obstacle}\right)=F_{req1}\left(C_{obstacle}\right)+F_{req2}\left(C_{goal}\right)$$(18)$$F_{req1}\left(C_{obstacle}\right)=K_{r}\left(\frac{1}{p\left(C_{arm},C_{obstacle}\right)}-\frac{1}{P_{obstacle}}\right)\frac{{p_{r}}^{n}}{p\left(C_{arm},C_{obstacle}\right)}$$(19)$$F_{req2}\left(C_{goal}\right)=\frac{n}{2}K_{r}{\left(\frac{1}{p\left(C_{arm},C_{obstacle}\right)}-\frac{1}{P_{obstacle}}\right)}^{2}\frac{{p_{r}}^{n-1}}{p\left(C_{arm},C_{obstacle}\right)}$$
where $F_{req1}\left(C_{obstacle}\right)$ the first component of the repulsive force, directed from the obstacle to the end actuator position of the robotic arm; $F_{req2}\left(C_{goal}\right)$ denotes the second component of the repulsive force, oriented from the end actuator position of the robotic arm toward the goal point.

By introducing a goal adjustment factor ${p_{r}}^{n}$, the repulsive field function gradually diminishes the repulsive force as the robot approaches the goal point. This approach effectively mitigates the issues of local minima and goal unreachability, thereby enhancing the efficiency and robustness of path planning.

#### 2.4.3. HP-RRT* Incorporating APF

In the implementation of the HP-RRT*, this paper employs an optimized artificial potential field method to incorporate obstacle perception information. The algorithm captures real-time spatial distribution information of obstacles and calculates the potential field value $U\left(S_{candidate}\right)$ for each candidate point to be integrated into the heuristic function. The selection of sampling points $S_{rand}$ are based on the probabilistic algorithm. Although this improves the directionality of the sampling process and accelerates the convergence of the algorithm, it does not fundamentally alter the randomness of the new node generation process. In complex environments, this may cause the algorithm to become trapped in a local minimum and experience deviations due to repeated sampling, thereby reducing the efficiency of the path search. In order to solve this problem, the RRT* algorithm combines heuristic probability sampling with the APF, referred to as HP-APF-RRT* for short. It utilizes the superposition of the gravitational potential field function $U_{att}\left(S_{goal}\right)$ associated with the target point and the repulsive potential field function $U_{req}\left(S_{obstacle}\right)$ related to the obstacle. This approach is applied to the neighboring nodes of the random tree algorithm $S_{nearest}$. Thus, the growth of random tree nodes is influenced not only by the gravitational potential field function $U_{att}\left(S_{nearest}\right)$ of the sampling point but also by the combined effects of target attraction and obstacle repulsion. Figure 9 illustrates the impact of the artificial potential field method on the expansion of random tree nodes.

The expressions for the gravitational potential field $U_{att}\left(S_{rand}\right)$ and the gravitational force $F_{rand}$ calculated for the neighboring node $S_{nearest}$ in the tree, with respect to the sampling point, are as follows:(20)$$U_{att}\left(S_{rand}\right)=\frac{1}{2}K_{c}p^{2}\left(S_{nearest},S_{rand}\right)$$(21)$$F_{att}\left(S_{rand}\right)=-\nabla U_{att}\left(S_{rand}\right)=K_{c}p\left(S_{nearest},S_{rand}\right)$$
where $U_{att}\left(S_{rand}\right)$ is the gravitational potential field at the $S_{rand}$; $K_{c}$ is the positively proportional gain coefficient.

The total potential field value is defined as the difference between the gravitational potential field and the repulsive potential field, expressed as:(22)$$U\left(S_{candidate}\right)=U_{att}\left(S_{goal}\right)-U_{req}\left(S_{obstacle}\right)$$

The expression for the heuristic function is denoted as:(23)$$f\left(n\right)=\left(e^{-\alpha \frac{dist\_to\_line}{radius}}\times e^{-\beta \frac{dist\_to\_b}{\left|\left|b-a\right|\right|+radius}}\right)+U\left(S_{candidate}\right)*\tau$$
where τ denotes the weighting factor used to balance the impact of $U\left(C_{arm}\right)$ on the total cost.

In the implementation process of the entire algorithm, weighted sampling in the sphere function is utilized to calculate the probability distribution, allowing for the random generation of the sampling point $S_{rand}$ in the configuration space. Subsequently, FindNearestPoint function is used to identify the node $S_{nearest}$ that is closest to $S_{rand}$ within the path planning tree. Next, the geometric angle θ between $S_{nearest}$ and $S_{rand}$, $S_{goal}$, and the set of obstacles $S_{obstacles}$ is computed using compute_angle function. Based on the known angle θ, the gravitational components $F_{rand}$ and $F_{goal}$ of $S_{nearest}$, $S_{rand}$, and $S_{goal}$ in the x, y and z axis directions, respectively, are calculated using compute_Attract function. At the same time, the repulsive force component $F_{req}$ of each obstacle relative to the nearest point $S_{nearest}$ is calculated in the x, y, and z axis directions, taking into account the compute repulsion and the angle θ. The combined force is determined by integrating the gravitational and repulsive components using the following expression:(24)$$F=F_{req}+F_{rand}+F_{goal}$$

The combined force determines the search direction, ω, according to the following expression:(25)$$\omega =\frac{F}{\sqrt{F{\left(1\right)}^{2}+F{\left(2\right)}^{2}+F{\left(3\right)}^{2}}}$$

The new node position, $S_{new}$, is calculated using Equation (26).(26)$$S_{new}=S_{nearest}+step*\omega$$

Perform collision detection on $S_{new}$ and add the path to the planning tree if no collisions are detected. The algorithm continues until the distance between the $S_{new}$ and the goal point is less than a specified threshold. Ultimately, it generates an optimal path from the starting point to the goal point. The general process of the algorithm is illustrated in Figure 10.

#### 2.4.4. Algorithm Time–Space Complexity Analysis

In this study, the time complexity of the HP-APF-RRT* algorithm is primarily influenced by the steps involved in candidate point generation, nearest node search, potential field calculation, and collision detection. In each iteration, the algorithm generates n candidate points and performs potential field calculations and collision detection for each candidate point. This results in a time complexity of O(n∗(N+M)) for each iteration, where N represents the number of nodes in the path planning tree and M denotes the number of obstacles. Overall, the time complexity of the algorithm is O(n∗T∗(N+M)), where T represents the number of iterations. By adopting a spatially partitioned data structure, the time complexity of the nearest node search can be optimized to O(logN) through the use of k-d trees, significantly reducing the overall time complexity.

In terms of space complexity, the HP-APF-RRT* algorithm must maintain the path planning tree, obstacle information, and a cache of n candidate points in each iteration. This results in a space complexity of O(n+N+M). The space requirements change dynamically with the algorithm; however, they are generally linearly related to the size of the data being processed by the algorithm.

### 2.5. Path Optimization Based on Trigonometric Inequalities

In three-dimensional space, pathfinding using the RRT* and its derivative algorithms produces a continuous line segment composed of a series of discrete points. However, this process frequently generates a significant number of redundant nodes. The presence of these nodes hinders the smooth operation of the robotic arm [36].

In this text, the triangle inequality is used to optimize each node in the path. Node cost evaluation is essential for parent node reselection, rewriting, and path optimization algorithms. Equation (27) is used to evaluate the trade-offs associated with node expansion by integrating path cost, security cost, and stability cost.(27)$$\cos t\left(c\right)=k_{1}P\left(c\right)+k_{2}\frac{1}{F\left(c\right)}+k_{3}T\left(c\right)$$
where $P\left(c\right)$ represents the cumulative path cost from the starting point to the current node c; $1/F\left(c\right)$ signifies the security cost, $F\left(c\right)$ is the reciprocal of the average distance between the current node c and its neighboring obstacles; $T\left(c\right)$ is the stability cost, which reflects the path depletion from the parent node to the current node c; $k_{1}$ $k_{2}$ $k_{3}$ serve as weighting factors that correspond to the respective sub-costs within the total cost calculation.

After obtaining the initial path, it undergoes further optimization. Refer to Figure 11 for a schematic representation of the optimization process. If the grandfather node of node C is used as its parent node, the cost can be reduced without causing a collision, meaning that the triangle inequality is satisfied. The parent node of node C is updated to its grandparent node. On the contrary, if the grandparent node of node C collides with node C, it is considered that both node C and its parent node have been optimized. This process is carried out recursively until the initial node is transformed into the node to be optimized, thereby completing the entire optimization process.

## 3. Experiments and Analysis

This section aims to experimentally validate the performance of the HP-APF-RRT*. For this reason, this study compares and analyzes the improved HP-APF-RRT* with the existing RRT, RRT*, P-RRT*, and the HP-RRT* proposed in this paper, all within a unified three-dimensional simulation environment. The goal is to verify the superiority, validity, and reliability of the improved HP-APF-RRT* in path planning.

The simulation experiments are conducted on a Windows 10 operating system platform equipped with an Intel Core i5-12490F processor (with a base frequency of 3.0 GHz) and 16 GB of RAM. An efficient, collision-free path from the initial point to the target point is successfully planned. In addition, this study also uses the NZ500-500 model manipulator from Jizhi Technology (Beijing, China) Co., Ltd. for simulation and physical experiments to further validate the practical application potential of the algorithm.

### 3.1. Simulation Experiment Analysis

In order to thoroughly assess the superiority of the algorithm proposed in this paper compared to existing algorithms, this study conducts comparative experiments within a fixed and challenging 3D spatial environment. All algorithms were tested through simulations in intricately designed 3D spatial environments.

In the experimental setup, the total area of the manipulable space for the robotic arm was established to be 1000 mm × 1000 mm. The gray areas on the map delineate the boundaries of the mobile space, and the cylinders, rectangles, and spheres within the map represent various types of obstacles. In each experiment, a consistent set of parameters is utilized for every algorithm. The experimental parameters are set as follows: the step size is set to 10 mm, the goal threshold is set to 50 mm, the search radius is set to 50 mm, and the maximum number of iterations is set to 10,000. In the P-RRT*, the probability of random sampling is set to 0.9.

In the HP-RRT* T and HP-APF-RRT* algorithms, the key parameters include the following: the number of candidate points (N), attenuation factors (*α*, *β*), the gravitational gain coefficient of the target point ($K_{a}$), the gravitational gain coefficient of the sampling point ($K_{c}$), the repulsive gain coefficient of the obstacle ($K_{r}$), the distance obstacle effect ($P_{obstacle}$), the gravitational distance constant ($p_{g}$), and the repulsive adjustment factor ($p_{r}$). The specific configuration of the parameters is shown in Table 2.

In order to ensure the statistical significance of the experimental results, 200 repetitive operations were conducted for each experimental group. In the three scenarios presented, the coordinates of the starting point in Scene 1 are (100, 100, 100) mm, while the coordinates of the target point are (900, 900, 900) mm. In Scene 2, the starting point remains at (100, 100, 100) mm, and the target point is located at (900, 900, 500) mm. In Scene 3, the starting point is again (100, 100, 100) mm, and the coordinates of the target point are (850, 850, 650) mm. The planning results of the algorithm are shown in Figure 12, Figure 13 and Figure 14. The average data from the experiments are systematically summarized in Table 3, Table 4 and Table 5. In parts (a) to (e) of the figure, the path graphs generated by various algorithms are presented. Among them, the blue line illustrates the exploration tree structure of the algorithm, and the red solid line denotes the path planned by the corresponding algorithm. In order to reflect the boundary extension property of the algorithm, no additional safety distances were incorporated into the actual collision judgment. As a result, the path lines appear to be very close to the surface of the obstacle.

In Scenario I, the performance of each algorithm demonstrates significant differences. The average search time of the RRT algorithm is 4.203 s, with an average of 5703 node samples, an average path length of 2112.441, and a search success rate of 91%. In contrast, the RRT* algorithm improves the path quality; however, the average search time increases to 7.262 s. The number of sampled nodes decreases slightly, while the success rate remains at 93%. The P-RRT* algorithm operates similarly to RRT*, exhibiting slightly different search times and path lengths while maintaining the same success rate. The HP-RRT* algorithm is significantly optimized for search time and node sampling while maintaining a 100% success rate and a short path length. The HP-APF-RRT* algorithm demonstrates optimal performance across all metrics, achieving an average search time of just 1.039 s, the fewest nodes sampled (2290), the shortest path length (1467.493), and a 100% success rate. Compared to RRT, HP-APF-RRT* offers approximately a 75% improvement in search time and a 30% reduction in path length.

In Scenario II, the average search time and the number of nodes sampled by the RRT algorithm increase to 4.625 s and 6304, respectively. Although the path length is slightly reduced, the success rate decreases to 73%. The RRT* algorithm excels in path optimization; however, the search time and the number of sampled nodes remain comparable, resulting in a significant decrease in the success rate to 61%. The P-RRT* algorithm experiences an increase in both search time and path length, accompanied by a modest rise in the success rate. The HP-RRT* algorithm continues to demonstrate advantages in search time and the number of nodes sampled. The path length is shorter, and the success rate is 100%. The HP-APF-RRT* algorithm outperforms all other methods across all metrics, achieving a search time of just 0.889 s, the fewest nodes sampled (1576), the shortest path length (1286.505), and a consistent success rate of 100%. Compared to RRT, HP-APF-RRT* improves search time by approximately 81% and reduces path length by about 33%.

In Scenario III, the average search time of the RRT algorithm is 2.241 s. The number of nodes sampled is 3984, the path length is 1880.449, and the success rate is 72%. The RRT* algorithm has an average path length of 1484.154. However, the search time increases to 3.674 s, the number of sampled nodes decreases, and the success rate remains the same as that of the RRT algorithm. The P-RRT* algorithm shows an increase in both search time and path length, along with a slight decrease in the success rate. The HP-RRT* algorithm offers advantages in search time and the number of nodes sampled, resulting in shorter path lengths and a 100% success rate. The HP-APF-RRT* algorithm completes its search in just 0.796 s, utilizing the fewest number of nodes sampled (1710) and finding the shortest path (1268.786) while maintaining a 100% success rate. Compared to RRT, HP-APF-RRT* reduces search time by approximately 65% and shortens path length by around 33%.

In order to verify the robustness and adaptability of the HP-APF-RRT* algorithm proposed in this paper within complex dynamic environments, two challenging dynamic test scenarios have been established. In both scenarios, the dynamic obstacle configurations encountered by each algorithm are kept consistent to ensure a fair evaluation of their performance. Additionally, the underlying configuration of the algorithm is consistent with previous experiments, ensuring the reliability and comparability of the experimental results. The results of the experiment are presented in Figure 15 and Figure 16. Figure 15 illustrates the various locations of obstacles and the effects of path planning for different algorithms, based on varying numbers of iterations in Dynamic Scenario 1. Figure 16 shows the different iterations of different algorithms, different positions of obstacles, and the effects of path planning in Dynamic Scene 2. The mean data from the experiments are systematically summarized in Table 6 and Table 7.

In Dynamic Scenario 1, the average search time of the HP-APF-RRT* algorithm is 0.439 s, making it 86.33% faster than the RRT algorithm. Additionally, the number of node samples is reduced by 84.57%, the path length is shortened by 29.46%, and the search success rate is 100%. In Dynamic Scenario II, the average search time of this algorithm is reduced to 0.380 s, which is 87.96% faster than the RRT algorithm, the number of node samples is reduced by 86.01%, the path length is shortened by 27.90%, and the search success rate is maintained at 100%. This demonstrates that the HP-APF-RRT* algorithm is both efficient and robust in dynamic environments, enabling it to quickly identify short and reliable paths.

In contrast, the RRT algorithm demonstrates consistent performance across both scenarios, with average search times of 3.187 and 3.133 s, node samples of 4596 and 4982, path lengths of 1950.186 and 1978.461 units, and search success rates of 76% and 85%. The RRT* algorithm has average search times of 7.59 and 6.811 s, node samples of 4952 and 5112, path lengths of 1508.926 and 1595.858 units, and search success rates of 82% and 84%. The P-RRT* algorithm has average search times of 5.487 and 5.988 s, with node samples of 4376 and 4778, path lengths of 1560.228 and 1571.750 units, and search success rates of 82% and 93%. The hp-RRT* algorithm exhibits average search times of 2.881 and 2.170 s, node sample counts of 2526 and 2337, path lengths of 1553.850 and 1513.380 units, and search success rates of 100%. Although the HP-RRT algorithm performs well, the HP-APF-RRT* algorithm offers greater advantages in terms of efficiency and robustness.

The experimental results show that the HP-APF-RRT* algorithm outperforms several key performance metrics, particularly in search time, sampling efficiency, path length, and search success rate. This improvement is primarily attributed to its heuristic sampling strategy and the incorporation of the artificial potential field method. These enhancements increase the algorithm’s sensitivity to obstacle perception and effectively direct the sampling points to circumvent obstacles. As a result, the number of required sampling points is significantly reduced, and the path optimization capability is enhanced. The HP-RRT* algorithm demonstrates improved performance; however, it is slightly less efficient than HP-APF-RRT* regarding search time and path length. The RRT, P-RRT*, and RRT* algorithms, although superior in certain aspects, fall short of the HP-APF-RRT* and HP-RRT in terms of overall performance. The RRT* algorithm, in particular, exhibits significant deficiencies in path optimization and robustness.

Finally, after optimizing the path in accordance with the triangular inequality, the optimization results presented in Table 8 have been obtained.

The optimization results indicate that in Scenario 1, the average path length was reduced from 1467.493 to 1461.463, achieving an optimization range of 0.4%. In Scenario 2, the average path length decreased from 1286.505 to 1275.741, resulting in an optimization range of 0.8%. In Scenario 3, the average path length improved from 1268.786 to 1261.857, with an optimization range of 0.5%. In Dynamic Scenario 1, the average path length was optimized from 1375.543 to 1350.293, yielding an optimization range of 1.8%. In Dynamic Scenario 2, the average path length was reduced from 1427.195 to 1396.321, achieving an optimization range of 2.2%. These data show that the optimized path length is reduced, further verifying the algorithm’s effectiveness in path optimization.

Through the experimental analysis, the HP-APF-RRT* proposed in this paper demonstrates superior performance in complex environments with multiple obstacles. In all scenarios, despite differing complexities, the HP-APF-RRT* achieves the shortest search time and the fewest number of node samples. It also provides the optimal path length and maintains a 100% search success rate. These results confirm that the HP-APF-RRT* surpasses other algorithms in search efficiency, node sampling efficiency, path optimization, and robustness, demonstrating its effectiveness and practicality in addressing complex path planning challenges.

### 3.2. Physical Experiment Analysis

In order to assess the effectiveness of the HP-APF-RRT* proposed in the text for practical applications, an experimental environment was established within the framework of the Robot Operating System (ROS) using the NZ500-500 model robotic arm from Jizhi Technology (Beijing, China) Co., Ltd. The initial step of the experiment was to construct a complex environmental scenario containing static obstacles, initial configuration points, and target configuration points within the Gazebo11 simulation software. In the ROS environment, the NZ500-500 manipulator model was loaded and visualized in three dimensions using the Gazebo tool. The HP-APF-RRT* directs the robotic arm using the MoveIt1 motion planning framework to facilitate obstacle avoidance. Ultimately, it successfully devises a collision-free path that satisfies the specified requirements.

As shown in Figure 17, key elements within the experimental scenarios have been designated with specific specialized symbols. The red square represents the target to be grasped, specifically the entity to be recognized and manipulated by the robotic arm. The purple columns symbolize the static obstacles in the path planning, which serve as physical barriers along the trajectory from the starting point to the endpoint. The black table represents the placement area, which is the final destination of the robotic arm’s operation and signifies the completion stage of the task.

Figure 17 illustrates the four key positions of the NZ500-500 robotic arm as it executes the path planning task from the starting point to the target point: the initial position, the gripping position, the intermediate position during movement, and the target position.

Real-time control of the NZ500-500 robotic arm and its interaction with the environment are achieved through the integration of the MoveIt motion planning framework with the ROS. Under this framework, the various key poses involved in the obstacle avoidance experiment of the robotic arm—including the initial, grasping, intermediate, and target poses—are precisely synchronized and scheduled using the topics and services of ROS. These intricate motion sequences and positional transformation processes are illustrated in Figure 18.

In order to verify the feasibility and superiority of the algorithm through real experiments, Under the same obstacle conditions described above, twenty simulations were conducted for each algorithm, and the results were averaged, as shown in Table 9.

The experimental data demonstrate that the HP-APF-RRT* exhibits high efficiency in motion planning during simulation experiments. This efficiency is evident in the algorithm’s optimality and completeness, specifically in its ability to determine the shortest paths and facilitate the fastest planning. The HP-APF-RRT* algorithm also performs well in physical experiments incorporating real-world application scenarios. Its performance satisfies the motion requirements of a real robotic arm, particularly in the time needed to execute the gripping action, which is shorter than that of other algorithms.

## 4. Conclusions

In this study, a six-degree-of-freedom robotic arm is selected as the research subject. The standard D–H parameter model is employed to establish its coordinate system and to derive both its forward and inverse kinematic equations. On this basis, an improved RRT* path planning algorithm (HP-APF-RRT*) combines a heuristic probability sampling strategy with the artificial potential field method. This approach aims to address the issues of low sampling efficiency, extended computation time, non-optimal paths, and failures in pathfinding within complex environments that are encountered by the traditional RRT* in three-dimensional space path planning. The algorithm minimizes redundant sampling nodes and enhances sampling efficiency by incorporating heuristic probabilistic sampling techniques. At the same time, the artificial potential field method is introduced to construct gravitational and repulsive fields. This approach guides the sampling points to avoid obstacles and move toward the target point, enhances the gravitational effect on the sampling points, and adjusts the direction of node extension. This method effectively addresses the pathfinding problem in complex environments. In addition, redundant intermediate nodes are streamlined using triangular inequalities to further optimize the path.

The results of the experimental simulation confirm the superiority of the algorithm compared to traditional algorithms, particularly in multi-obstacle sampling environments. Physical experiments further validate the feasibility of the algorithm in practical applications. Compared to the traditional RRT* algorithm, the running speed of this algorithm has been significantly enhanced.

The work presented in this paper has significant potential for improvement. Real-time path planning for dynamic target points is a significant research challenge, particularly in the context of utilizing reinforcement learning algorithms for moving target points. In real-world applications, such as robotic arm garbage sorting, real-time target tracking, and dynamic target path planning, traditional path planning algorithms must be integrated with reinforcement learning. This combination enhances adaptability and robustness in response to dynamically changing environments.

## Figures and Tables

**Figure 1 sensors-25-00328-f001:**
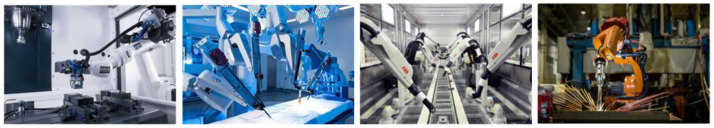
Industrial robot.

**Figure 2 sensors-25-00328-f002:**
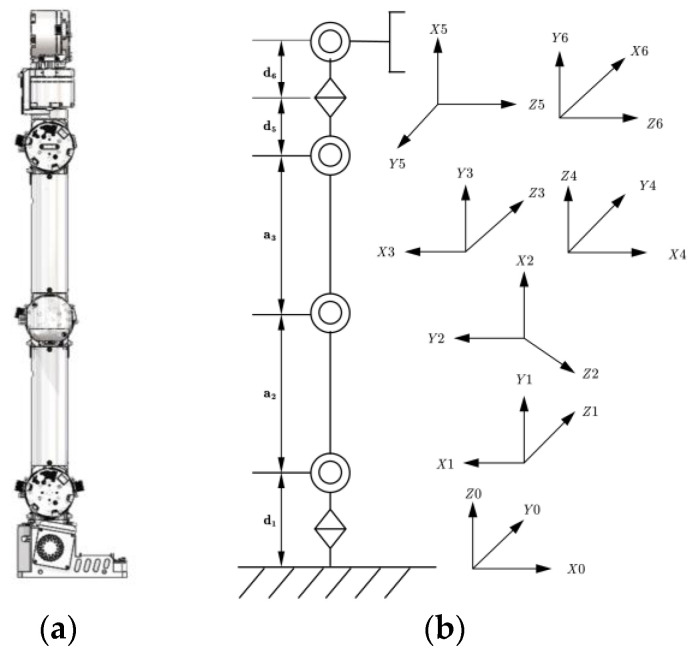
D–H coordinate system of the six-degree-of-freedom manipulator: (**a**) 3D model, (**b**) structural diagram of the manipulator arm, and D–H coordinate system.

**Figure 3 sensors-25-00328-f003:**
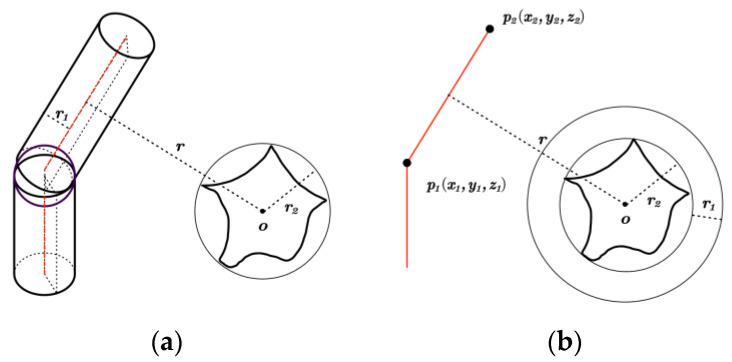
Collision detection model of a robotic arm with irregular obstacles: (**a**) original collision model; and (**b**) simplified collision model.

**Figure 4 sensors-25-00328-f004:**
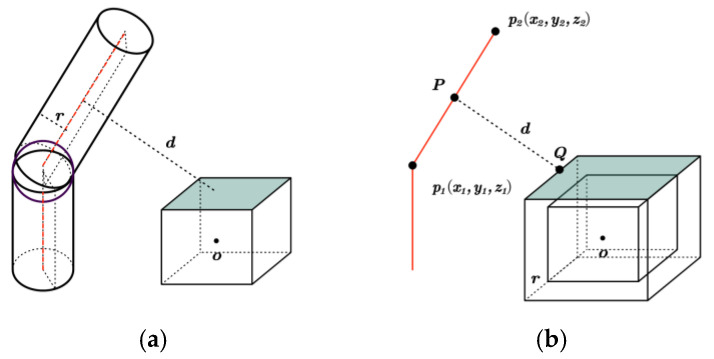
Collision detection model of a robotic arm with regular obstacles: (**a**) original collision model; and (**b**) simplified collision model.

**Figure 5 sensors-25-00328-f005:**
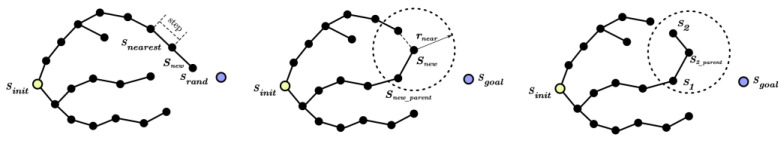
RRT* algorithm flow.

**Figure 6 sensors-25-00328-f006:**
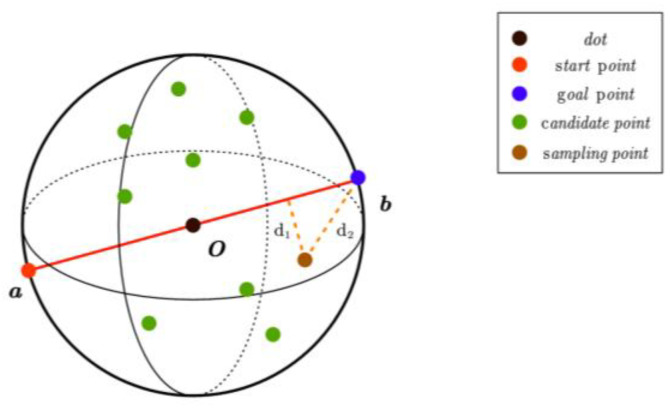
Sampling space.

**Figure 7 sensors-25-00328-f007:**
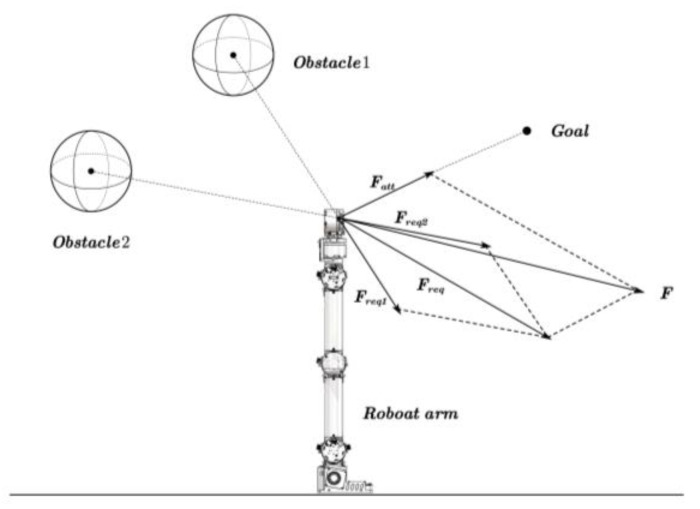
Robotic arm obstacle avoidance model of APF algorithm.

**Figure 8 sensors-25-00328-f008:**
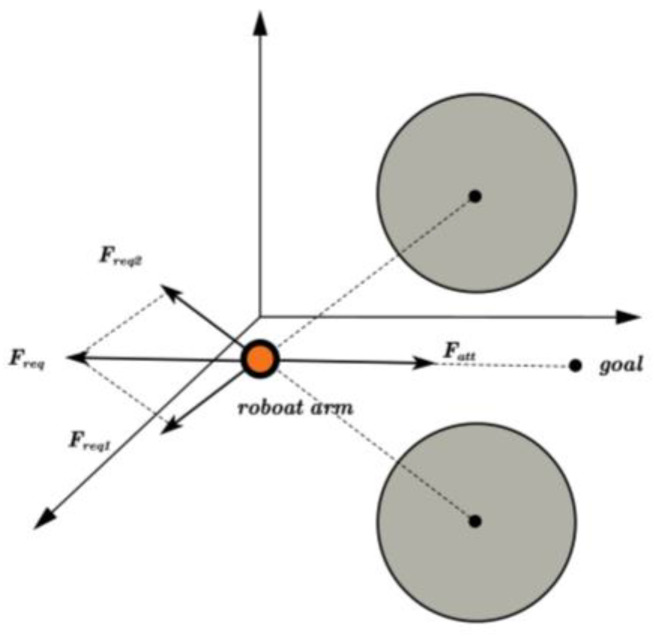
Potential well problem.

**Figure 9 sensors-25-00328-f009:**
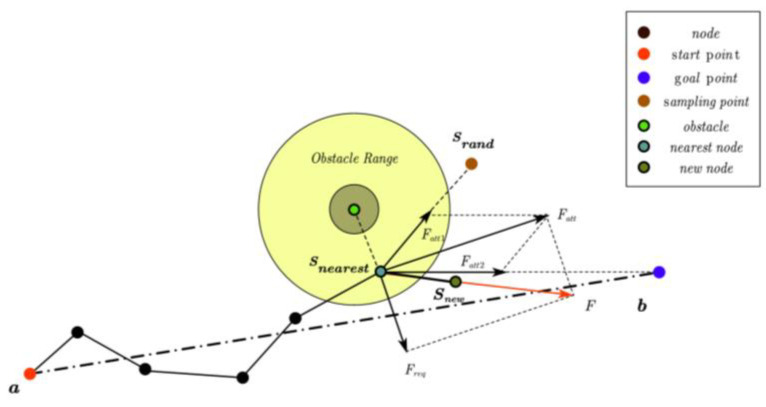
Schematic diagram of the algorithm.

**Figure 10 sensors-25-00328-f010:**
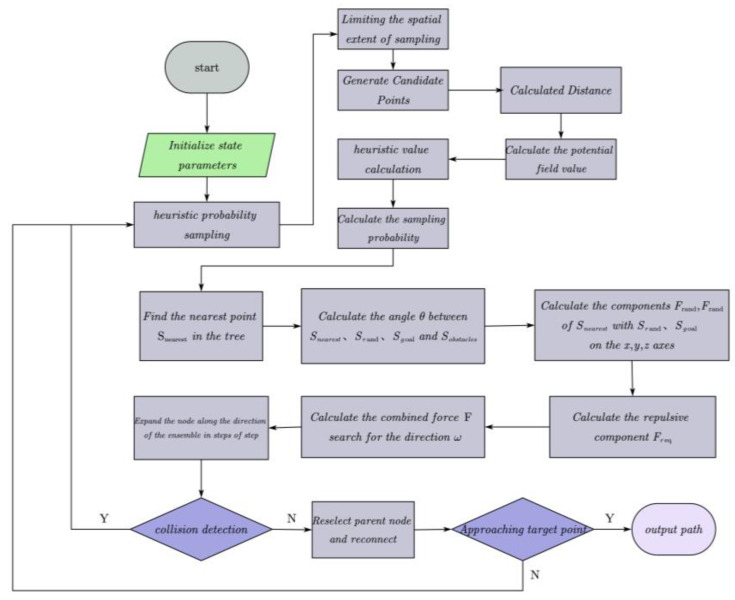
Flowchart of the algorithm.

**Figure 11 sensors-25-00328-f011:**
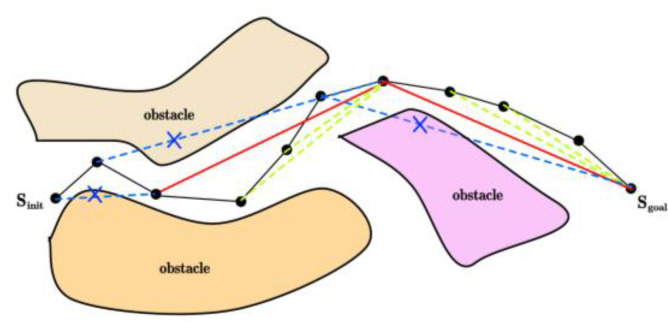
Path optimization.

**Figure 12 sensors-25-00328-f012:**
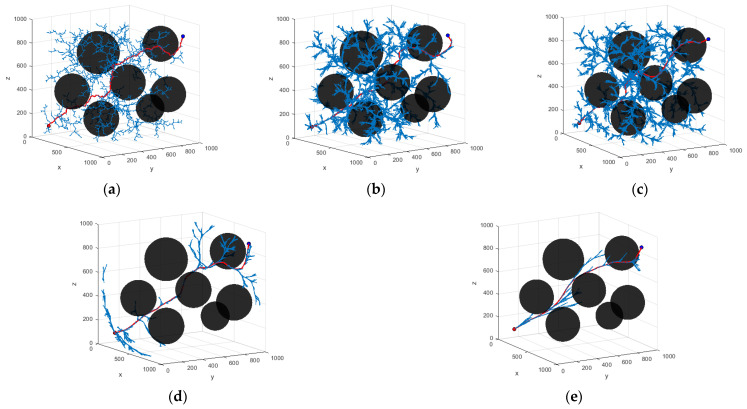
Different algorithms to generate paths in Scenario I: (**a**) RRT; (**b**) RRT*; (**c**) P-RRT*; (**d**) HP-RRT*; and (**e**) HP-APF-RRT*.

**Figure 13 sensors-25-00328-f013:**
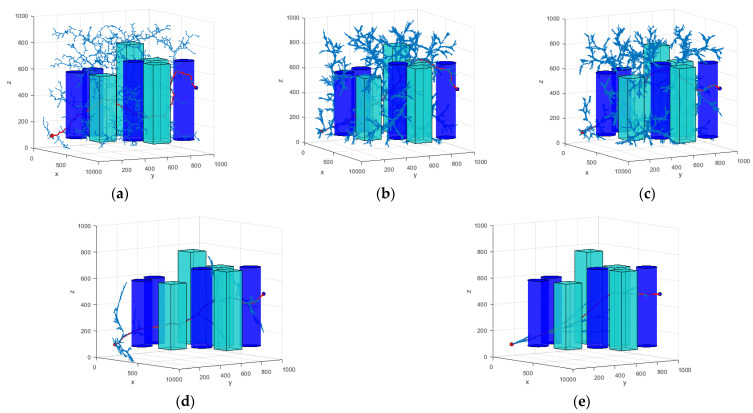
Different algorithms to generate paths in Scenario II: (**a**) RRT; (**b**) RRT*; (**c**) P-RRT*; (**d**) HP-RRT*; and (**e**) HP-APF-RRT*.

**Figure 17 sensors-25-00328-f017:**
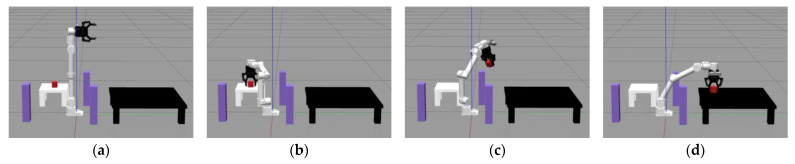
Robotic arm poses: (**a**) initial pose; (**b**) gripping pose; (**c**) intermediate pose; and (**d**) target pose.

**Figure 18 sensors-25-00328-f018:**
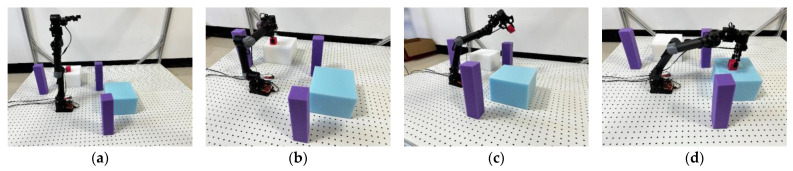
Real robotic arm poses: (**a**) initial pose; (**b**) gripping pose; (**c**) intermediate pose; and (**d**) target pose.

**Figure 14 sensors-25-00328-f014:**
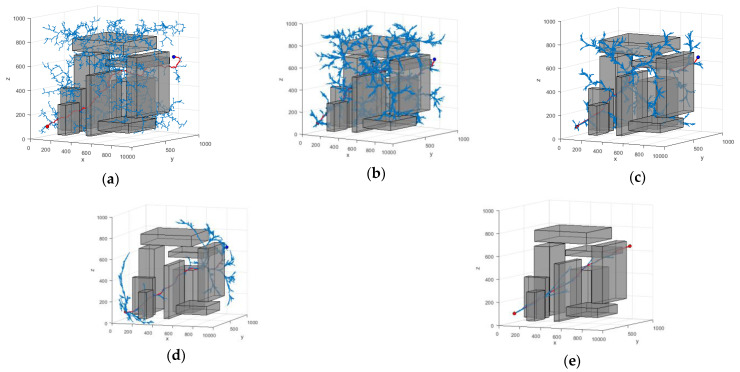
Different algorithms to generate paths in Scenario III: (**a**) RRT; (**b**) RRT*; (**c**) P-RRT*; (**d**) HP-RRT*; and (**e**) HP-APF-RRT*.

**Figure 15 sensors-25-00328-f015:**
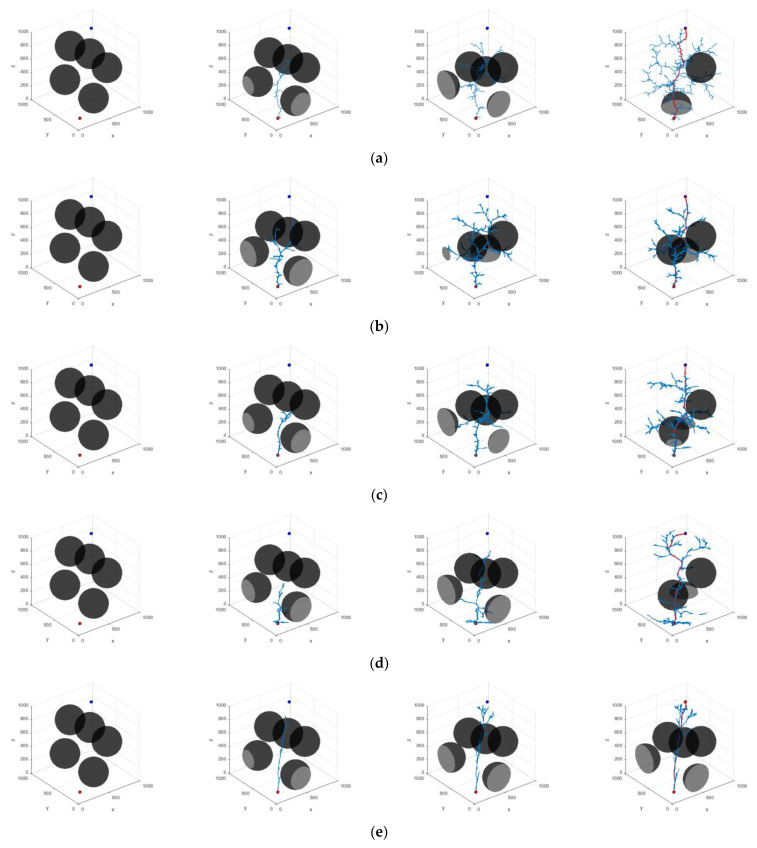
Different algorithms to generate paths in Dynamic Scene I: (**a**) RRT; (**b**) RRT*; (**c**) P-RRT*; (**d**) HP-RRT*; and (**e**) HP-APF-RRT*.

**Figure 16 sensors-25-00328-f016:**
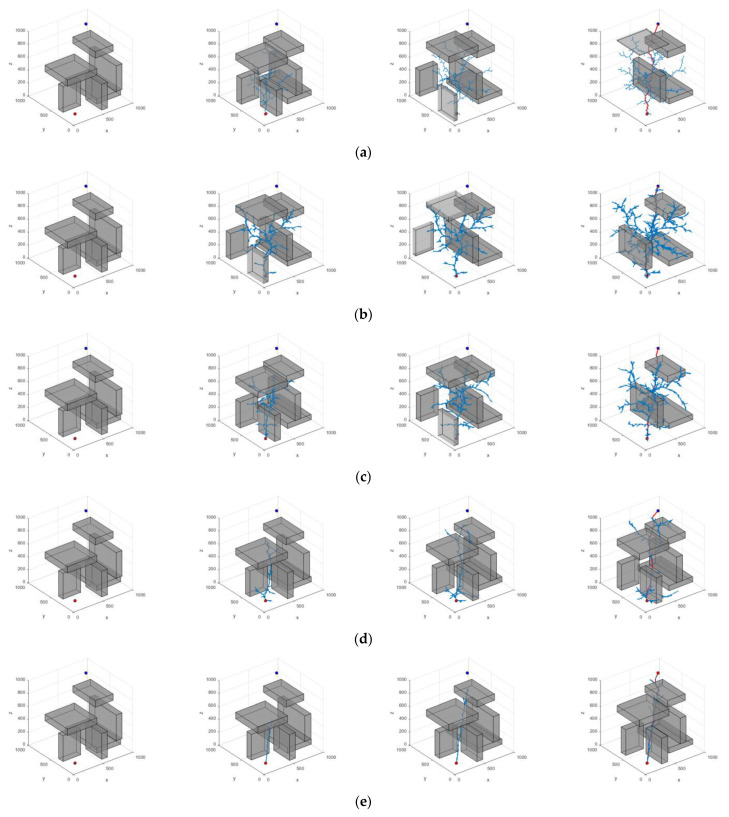
Different algorithms to generate paths in Dynamic Scene II: (**a**) RRT; (**b**) RRT*; (**c**) P-RRT*; (**d**) HP-RRT*; and (**e**) HP-APF-RRT*.

**Table 1 sensors-25-00328-t001:** Standard D–H parameters of six degrees of freedom manipulator.

Connect Rod Serial Number (i)	$a_{i}$/mm	$\alpha_{i}$/(°)	$d_{i}$/mm	$\theta_{i}$ (°)	$\beta_{i}$ (°)
1	0	−90	92.5	$\theta_{1}$	±180
2	189	90	0	$\theta_{2}$	±135
3	189	−90	0	$\theta_{3}$	±150
4	0	0	0	$\theta_{4}$	−170~+180
5	0	−90	36	$\theta_{5}$	±120
6	0	0	86	$\theta_{6}$	±360

**Table 2 sensors-25-00328-t002:** Key parameters.

Parameter	N	*α*	*β*	$K_{a}$	$K_{c}$	$K_{r}$	$P_{obstacle}$	$p_{g}$	$p_{r}$
No. of Equation	—	(5)	(5)	(14)	(20)	(16)	(16)	(14)	(16)
Value	10	0.6	0.4	1.5	1	1	50	300	1.2

**Table 3 sensors-25-00328-t003:** Comparison of Scene I algorithms.

Algorithm Name	Average Search Time/s	Average Number of Node Samples	Average Path Length	Search Success Rate
RRT	4.203	5703	2112.441	91%
RRT*	7.262	5513	1686.495	93%
P-RRT*	7.768	5539	1690.284	93%
HP-RRT*	3.015	2544	1713.812	100%
HP-APF-RRT*	1.039	2290	1467.493	100%

**Table 4 sensors-25-00328-t004:** Comparison of Scene II algorithms.

Algorithm Name	Average Search Time/s	Average Number of Node Samples	Average Path Length	Search Success Rate
RRT	4.625	6304	1935.640	73%
RRT*	10.415	6297	1530.290	61%
P-RRT*	11.925	6488	1549.293	67%
HP-RRT*	2.237	1978	1445.297	100%
HP-APF-RRT*	0.889	1576	1286.505	100%

**Table 5 sensors-25-00328-t005:** Comparison of Scene III algorithms.

Algorithm Name	Average Search Time/s	Average Number of Node Samples	Average Path Length	Search Success Rate
RRT	2.241	3984	1880.449	72%
RRT*	3.674	3603	1484.154	72%
P-RRT*	5.168	4200	1498.405	70%
HP-RRT*	2.047	1941	1449.738	100%
HP-APF-RRT*	0.796	1710	1268.786	100%

**Table 9 sensors-25-00328-t009:** Comparison of algorithms.

Algorithm Name	Average Grasp Time/s	Average Search Time/s	Search Success Rate
RRT	49.65	24.75	85%
RRT*	58.74	36.71	80%
P-RRT*	53.61	32.95	85%
HP-RRT*	24.85	7.56	100%
HP-APF-RRT*	19.79	5.62	100%

**Table 6 sensors-25-00328-t006:** Comparison of Dynamic Scene I algorithms.

Algorithm Name	Average Search Time/s	Average Number of Node Samples	Average Path Length	Search Success Rate
RRT	3.187	4596	1950.186	76%
RRT*	7.59	4952	1508.926	82%
P-RRT*	5.487	4376	1560.228	82%
HP-RRT*	2.881	2526	1553.850	100%
HP-APF-RRT*	0.439	703	1375.543	100%

**Table 7 sensors-25-00328-t007:** Comparison of Dynamic Scene II algorithms.

Algorithm Name	Average Search Time/s	Average Number of Node Samples	Average Path Length	Search Success Rate
RRT	3.133	4982	1978.461	85%
RRT*	6.811	5112	1595.858	84%
P-RRT*	5.988	4778	1571.750	93%
HP-RRT*	2.170	2337	1513.380	100%
HP-APF-RRT*	0.380	690	1427.195	100%

**Table 8 sensors-25-00328-t008:** Comparison of optimization algorithms.

Algorithm Name	Average Path Length	Average Optimized Path Length
Scenario I	1467.493	1461.463
Scenario II	1286.505	1275.741
Scenario III	1268.786	1261.857
Dynamic Scene I	1375.543	1350.293
Dynamic Scene II	1427.195	1396.321

## Data Availability

The data presented in this study are available on request from the corresponding author.

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
