# Peer review of "Path Planning Algorithm for Manipulators in Complex Scenes Based on Improved RRT*"

_sensors, 2025, doi:10.3390/s25020328_

Round 1

Reviewer 1 Report

Comments and Suggestions for Authors

The paper proposes an improved fast extended random tree algorithm for robotic arm path planning method (HP-APF-RRT) that combines heuristic probability sampling and artificial potential field methods, which performs well in complex environments. The paper provides a detailed theoretical analysis and verifies the effectiveness and superiority of the algorithm through simulation and physical experiments. The paper is generally excellent and the work is relatively complete, but there are still areas for improvement. For example, although the paper conducted experiments in two scenarios, adding more different types of scenarios (such as dynamic obstacles) can further verify the robustness of the algorithm. In addition, there is relatively little analysis of the time and space complexity of the algorithm in the paper, and more discussion can be added in this regard to better understand the performance of the algorithm.

Reviewer 2 Report

Comments and Suggestions for Authors

The manuscript is fine but lacks novelty and is not original.

The authors need to explain what parameters have been used for the algorithms, and the rationale behind it. 

What are the length of tree edges, the sample size before timing out, and the probability of checking for a connection to the goal, are these values consistent across the iterations for various algorithms used in Table 2.

What is the percentage improvement? It is recommended to use more scenarios.

There are some grammatical issues, the English needs to be improved,
